# The structural and functional characterization of human RecQ4 reveals insights into its helicase mechanism

Sebastian Kaiser[1], Florian Sauer[1] & Caroline Kisker[1]

RecQ4 is a member of the RecQ helicase family, an evolutionarily conserved class of enzymes, dedicated to preserving genomic integrity by operating in telomere maintenance, DNA repair and replication. While reduced RecQ4 activity is associated with cancer predisposition and premature aging, RecQ4 upregulation is related to carcinogenesis and metastasis. Within the RecQ family, RecQ4 assumes an exceptional position, lacking several characteristic RecQ domains. Here we present the crystal structure of human RecQ4, encompassing the conserved ATPase core and a novel C-terminal domain that lacks resemblance to the RQC domain observed in other RecQ helicases. The new domain features a zinc-binding site and two distinct types of winged-helix domains, which are not involved in canonical DNA binding or helicase activity. Based on our structural and functional analysis, we propose that RecQ4 exerts a helicase mechanism, which may be more closely related to bacterial RecQ helicases than to its human family members.

[1] Rudolf-Virchow-Center for Experimental Biomedicine, Institute of Structural Biology, Josef-Schneider-Str. 2/D15, Wuerzburg 97080, Germany. Correspondence and requests for materials should be addressed to C.K. (email: Caroline.Kisker@Virchow.uni-wuerzburg.de).

Preservation of genomic integrity is a prime objective for every living organism to maintain cell homeostasis and to pass on conserved genetic information to the following generation. The family of RecQ helicases has emerged as a vital class of enzymes to ensure proper genome maintenance by performing a broad variety of functions in DNA replication, recombination and repair as well as telomere stability[1,2]. RecQ proteins are highly conserved from bacteria to humans and impaired function leads to increased incidences of genomic fragility, chromosomal breakage and rearrangements, ultimately promoting carcinogenesis and accelerated aging in higher organisms. Intriguingly, overexpression of RecQ helicases may also promote tumorigenesis and RecQ4 in particular is highly upregulated in cervical, prostate and breast cancers[3–5]. Generally, one RecQ homologue is found in single cellular organisms, whereas higher life forms express multiple paralogues. On the basis of homology to the ATPase core domain of the founding member in *Escherichia coli* (*Ec*RecQ), five RecQ helicases have been identified in humans: RecQ1, BLM, WRN, RecQ4 and RecQ5. While RecQ1 and RecQ5 have not been linked to human pathogenesis, defects in BLM and WRN are causative for Bloom Syndrome[6] and Werner Syndrome[7], respectively. RecQ4 mutations are linked to three autosomal recessive diseases: Rothmund-Thomson-Syndrome (RTS) type II, RAPADILINO Syndrome and Baller-Gerold-Syndrome (BGS). All three RecQ4-associated syndromes share common clinical features including skeletal abnormalities (for example, radial ray defects) and growth retardation. In addition, RTS type II individuals are characterized by skin abnormalities (poikiloderma), symptoms of premature aging and a high risk for developing osteosarcoma. While the RAPADILINO phenotype typically lacks the unique RTS characteristics, these patients feature an elevated risk for developing both, osteosarcoma and lymphoma[8].

RecQ proteins are ATP- and DNA-dependent super family 2 (SF2) helicases and separate double stranded DNA (dsDNA) in a 3′ to 5′ direction. Most RecQ helicases feature a set of conserved domains, thereby permitting genuine helicase function, interactions with other proteins and, most importantly, the interaction with a variety of DNA-metabolism intermediate structures. These RecQ-characteristic domains are the ATPase motor domain, the RecQ-C-terminal (RQC) domain and the Helicase and RNase D-like C-terminal domain (HRDC; Fig. 1a). The most conserved element is the ATPase core, comprising two tandem RecA-like folds termed helicase domain 1 and 2 (HD1 and HD2), which feature the characteristic sequence motifs of SF2 helicases[9]. In addition, the HD1 of all RecQ helicases features a conserved aromatic-rich sequence motif, the aromatic-rich loop (ARL), located between helicase motifs II and III[10]. This motif represents a ssDNA sensor element, coupling DNA-binding to structural rearrangements in HD1, thereby indirectly enabling ATP hydrolysis. Less conserved, the RQC domain is located downstream of the HDs and, in concert with the ATPase domain, constitutes the functional core unit for helicase activity. The RQC domain consists of a $Zn^{2+}$-binding element and a winged-helix (WH)-like domain, which mediates sequence-independent contacts with the DNA and is further involved in the oligomerization status of RecQ helicases[11,12]. In addition, a β-hairpin element in the RecQ-conserved WH domain was found to destabilize the Watson–Crick base-pairing at the ssDNA/dsDNA junction and thereby enables helicase function[11–14]. Although the WH domain is conserved in most RecQ proteins, the structural importance of the β-hairpin for dsDNA separation was demonstrated only for the human RecQ helicases RecQ1 (refs 11,12), WRN[13] and BLM[14]. Bacterial RecQ homologues contain a shorter version of the β-hairpin, which does not interact with the bases at the

ssDNA/dsDNA junction. This observation led to the proposal that bacterial RecQs might utilize a helicase mechanism, which is based on a critical bend angle instead of a β-hairpin element to destabilize base pairing[15]. Remarkably, RecQ4 entirely lacks the RQC domain, yet its helicase activity is required for various DNA protective functions[16,17]. This raises the question of how dsDNA separation is achieved in this important RecQ homologue. The third and least conserved structural element of the RecQ helicase family is the HRDC domain, which is only present in BLM and WRN as well as in bacterial and yeast homologues[18]. The low sequence conservation of HRDC domains results in distinct surface properties, which were suggested to mediate unique protein interactions of individual RecQ helicases, fine-tuning their distinctive cellular functions[19,20].

The human *recq4* gene encodes a 1208 amino acid (aa) polypeptide, resulting in a 133 kDa protein with the highly conserved ATPase domain located in its center[21] (Fig. 1a). The N-terminal region possesses homology to the essential DNA replication initiation factor Sld2 from *Saccharomyces cerevisiae* (aa 1–400)[22,23]. Accordingly, RecQ4 assumes an important function in the initiation of eukaryotic DNA replication[22,24,25]. In addition to a nuclear targeting signal (NTS)[26], a mitochondrial localization sequence was identified within the N-terminus[27]. Thus, RecQ4 is the only RecQ helicase found in mitochondria as well as in the cytosol and the nucleus, raising the possibility that RecQ4 is also involved in mtDNA replication and/or maintenance[28,29]. Although the N-terminus is predicted to be largely unstructured, specific binding regions for RNA, Holiday-Junctions and G-quadruplex DNA have been identified throughout the entire N-terminal 430 aa (refs 30–32). While the N-terminus is indispensible for viability due to its implication in DNA replication initiation and embryogenesis[33], the DNA protective character of RecQ4 has been assigned to its helicase domain and the unique C-terminus[34,35] and the vast majority of the disease-causing mutations within RecQ4 are located in these regions[8]. However, high-resolution structural information was so far limited to two small regions within the N-terminal part of RecQ4. NMR studies of the first 54 aa revealed a homeodomain-like DNA-interaction motif[36] and more recently, a short $Zn^{2+}$-binding motif upstream of the ATPase domain, the so-called Zn-knuckle (aa 610–634), was characterized in *Xenopus laevis* RecQ4 (ref. 31).

Here we present the crystal structure of the human RecQ4 helicase, encompassing its central ATPase domain and a large fraction of its unique C-terminus. Downstream of the ATPase core, we identify a novel domain, which features homology to WH domains of various prokaryotic DNA-binding proteins and coordinates a $Zn^{2+}$-ion, yet lacks structural resemblance to the conserved RQC domain. Our structural and biochemical analysis provides insights into DNA substrate binding by RecQ4 and its helicase mechanism and allows us to draw conclusions about the functional relevance of RecQ4-associated diseases.

## Results

**Overall structure**. The RecQ4[427-1116] model was refined to a resolution of 2.75 Å (Table 1) and contains aa 449–1111 (Fig. 1b). Two disordered regions within the C-terminus (aa 858–882 and aa 975–983) and two short areas within the ATPase domain (aa 664–667 and aa 727–734) are not resolved. HD1 and HD2 adopt the typical RecA-like fold, which is characteristic for SF1 and SF2 helicases[9]. Two 'bridging' α-helices (aa 823–835 and aa 1046–1079) link HD2 to a unique domain further downstream (aa 836–1045), which is not present in other RecQ helicases. One feature of this novel domain is the coordination of a $Zn^{2+}$-ion (Supplementary Fig. 3 and Table 1). This new domain (subsequently designated RecQ4-$Zn^{2+}$-binding domain—

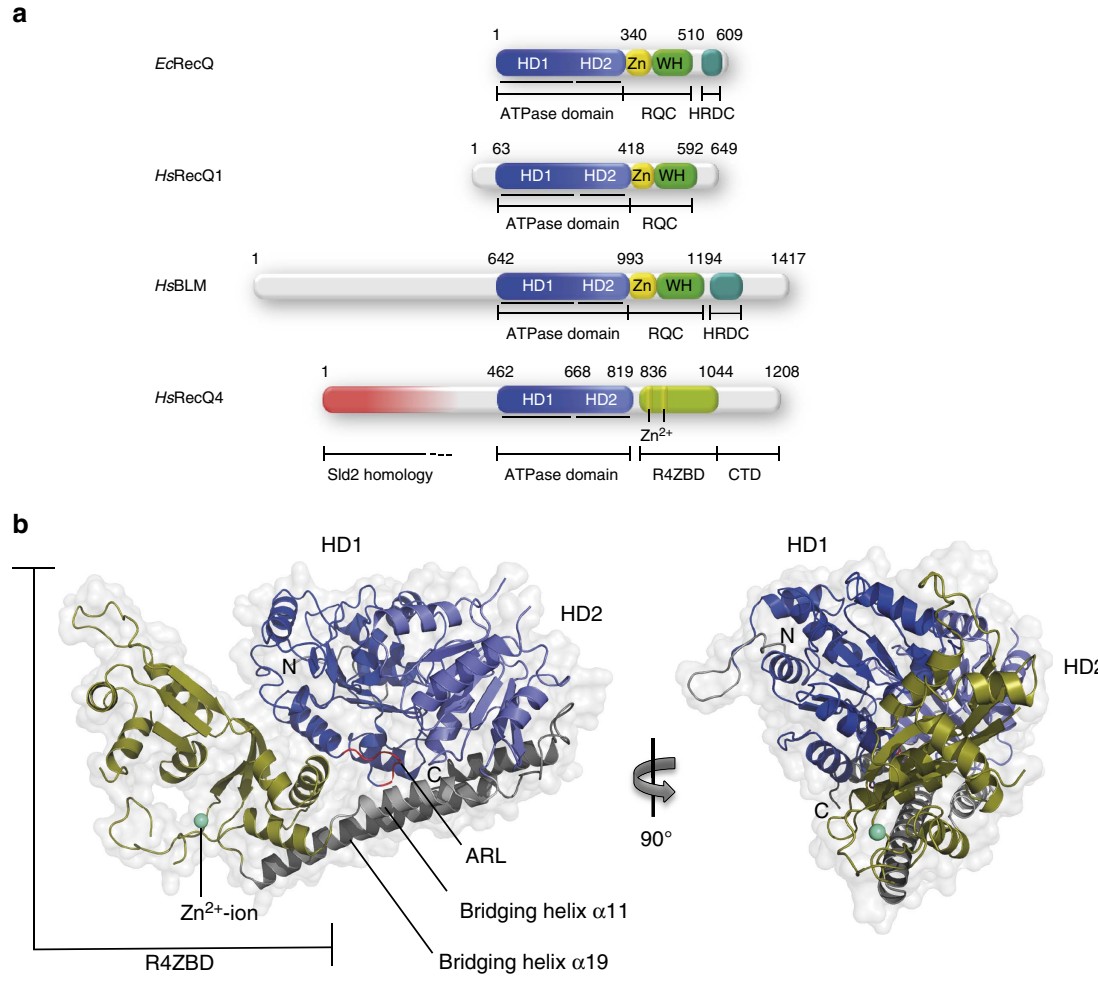

**Figure 1 | Overall structure of RecQ4.** (**a**) Comparison of the domain architecture of *E. coli* (*Ec*)RecQ and the *Homo sapiens* (*Hs*)RecQ helicases BLM, RecQ1 and RecQ4. The ATPase domain, comprising HD1 and HD2, is highly conserved among all RecQ proteins. Additional RecQ-conserved domains are the RQC domain (featuring the $Zn^{2+}$-binding (Zn) and winged-helix (WH) subdomains) and the HRDC domain. In place of a RQC domain, RecQ4 features a structurally unique domain, termed RecQ4-$Zn^{2+}$-binding domain (R4ZBD). Upstream of the helicase core, RecQ4 harbors the Sld2-homology domain at its N-terminus. Colour code: HD1—dark blue, HD2—light blue, Zn—yellow, WH—green, HRDC—turquoise, Sld2—red, R4ZBD—olive. (**b**) Structure of human RecQ4 (aa 449–1111) in cartoon representation. Colour code as in A. Two bridging helices, α11 and α19 connect the ATPase domain to the R4ZBD in a hinge-like fashion. The R4ZBD coordinates a $Zn^{2+}$-ion (cyan sphere). The position of the ARL within HD1 is indicated and shown in red. N- and C-termini are indicated in both structures by the letters N and C, respectively. A complete annotation of secondary structure elements is provided in Supplementary Fig. 1. A stereo-view image of the R4ZBD including its electron density is shown in Supplementary Fig. 2.

R4ZBD) can be separated into two halves (Fig. 2a). The lower half consists of three antiparallel β-strands and three short α-helices. $Zn^{2+}$-coordination is accomplished via three cysteines and a histidine, which are located in the loop region between helix α12 and β-strand β13. The upper half of the R4ZBD features three antiparallel β-strands, which are packed against a four-helix bundle. Sequentially, the upper half of the R4ZBD represents an insertion within the lower half (Fig. 2b). Thus, the entire R4ZBD does not constitute two autonomous domains as observed for the RQC domains of other RecQ family members, which feature an individual $Zn^{2+}$-binding followed by a WH-subdomain[12,13,14,37]. Following the R4ZBD, the polypeptide chain transits into the second bridging helix (aa 1046–1079), leading back towards HD2. The last helix in the model, helix α20 (aa 1094–1109), folds back onto the downstream section of the large bridging helix and both α-helices pack tightly against HD2. The two bridging helices thus connect the HD2 to the R4ZBD in a hinge-like fashion, with the joint of the hinge being located at the HD2–bridge–helix interface. The R4ZBD, on the other side of this lever, interacts with HD1 through several salt bridges and hydrogen bonds.

**The R4ZBD features homology to winged-helix domains**. To identify the functional role of the newly identified R4ZBD we subjected the R4ZBD including the proximal half of bridging helix α19 (aa 836–1060) to a DALI search against the PDB data base[38]. Overall, 246 homologous structures were identified. The vast majority of these structures (221/246) mapped a homology exclusively to the lower half of the R4ZBD. The ten most similar structures comprise nine bacterial transcription factors and a DNA-replication initiation factor from *E. coli* (Supplementary Table 1). All these proteins bind dsDNA via a WH-like DNA-binding motif and their homology towards the lower half of the R4ZBD is consistent with these DNA-binding domains (Fig. 2c, left). We further subjected the coordinates of the isolated upper half (aa 942–1032) to a DALI PDB search, however, compared to the entire R4ZBD, the outcome was less consistent regarding the function of these structural homologues (Supplementary Table 2). The closest homologue for the isolated upper half is the human methionine aminopeptidase 2 (Fig. 2c, right). Notably, neither of the identified structural homologues for the complete R4ZBD nor for its upper half included a WH domain from other RecQ helicases.

**Table 1 | Data collection and refinement statistics.**

| | High resolution* | SAD Phasing* | Zn Detection† | |
|---|---|---|---|---|
| *Data collection* | | | | |
| Space group | $P3_121$ | $P3_121$ | $P3_121$ | |
| Cell dimensions | | | | |
| $a$, $b$, $c$ (Å) | 131.00, 131.00, 96.34 | 130.06, 130.06, 95.84 | 128.96, 128.96, 94.68 | 129.19, 129.19, 94.93 |
| $\alpha$, $\beta$, $\gamma$ (°) | 90, 90, 120 | | | |
| | | | Peak | Low energy |
| Wavelength | 0.9763 | 1.2828 | 1.2823 | 1.2831 |
| Resolution (Å) | 44.34–2.75 (2.90–2.75) | 48.56–2.91 (3.09–2.91) | 48.10–3.00 (3.18–3.00) | 48.20–3.00 (3.18–3.00) |
| $R_{pim}$ | 1.9 (51.6) | 2.3 (27.2) | 2.5 (27.3) | 2.4 (34.0) |
| CC1/2 | 99.9 (76.0) | 100 (87.0) | 99.9 (94.9) | 100 (100) |
| $I/\sigma I$ | 21.7 (1.4) | 25.0 (3.1) | 26.0 (4.1) | 27.9 (3.5) |
| Completeness (%) | 99.8 (100) | 99.5 (97.2) | 100 (100) | 100 (100) |
| Redundancy | 14.3 (14.3) | 19.6 (20.1) | 19.8 (20.9) | 19.9 (20.9) |
| | | | | |
| *Refinement* | | | | |
| Resolution (Å) | 44.34–2.75 | | | |
| No. reflections (free) | 47984 (2459) | | | |
| $R_{work}/R_{free}$ (%) | 18.3/23.1 | | | |
| No. atoms | | | | |
| Protein | 4692 | | | |
| $Zn^{2+}$ | 1 | | | |
| *B*-factors | | | | |
| Protein | 108.8 | | | |
| $Zn^{2+}$ | 83.1 | | | |
| r.m.s deviations | | | | |
| Bond lengths (Å) | 0.004 | | | |
| Bond angles (°) | 0.681 | | | |

r.m.s., root mean square; SAD, single-wavelength anomalous dispersion.
*High resolution and SAD data sets were recorded on individual crystals.
†Peak- and low-energy data sets were recorded on the same crystal.

**Comparison to other RecQ structures**. We superimposed our RecQ4 structure with the two human paralogues RecQ1 and BLM, as well as the *E. coli* RecQ (*Ec*RecQ) structure (PDB entries 2wwy: RecQ1, 4o3m: BLM, 1oyw: *Ec*RecQ). The structurally most conserved domain of all RecQ helicases is the second RecA-like domain (HD2), which therefore served as the basis for the superposition (Fig. 3). Although structurally well conserved, HD1 aligns less well and adopts a range of different orientations towards HD2 in the individual structures, highlighting the inherent level of flexibility of both RecA-like domains relative to each other. The R4ZBD of RecQ4 is located directly adjacent to HD1 (Fig. 3a) and thus assumes a unique position among the characterized RecQ proteins, as their C-terminal extensions are located on the opposite side, directly adjacent to HD2 (Fig. 3b–d).

The $Zn^{2+}$-binding domain within the RQC domain of other RecQ helicases is positioned at the interface to HD2, sandwiched between HD2 and the WH domain. It consists of four α-helices, with the last two α-helices arranged in-line in an antiparallel order, coordinating the $Zn^{2+}$-ion between them[12,13,14,37]. Interestingly, the exact position of the RQC-$Zn^{2+}$-binding domain is occupied by segments of RecQ4s bridging helices and the further downstream helix α20 (Fig. 4). While the bridging helix α11 and the downstream segment of bridging helix α19 in RecQ4 cover the non-$Zn^{2+}$-coordinating helices of the RQC-$Zn^{2+}$-binding domain, helix α20 assumes the location of the two shorter in-line oriented helices, which coordinate the metal ion. Notably, the mode of $Zn^{2+}$ coordination by the R4ZBD diverges substantially from the mode of $Zn^{2+}$ coordination of the RQC domains. While the $Zn^{2+}$-ion in RQC domains is exclusively coordinated via four cysteines within the last two helices of the RQC-$Zn^{2+}$-binding domain, the R4ZBD coordinates the $Zn^{2+}$-ion via three cysteines and a histidine, all of which are located in a loop region, connecting α12 and β13 (Fig. 2a).

**The CTD of RecQ4 is critical for helicase activity**. The functional core unit of RecQ helicases consists of the ATPase- and the RQC domain. Especially the WH subdomain is indispensable for helicase activity[11,12,14,39]. As the RQC domain is missing in RecQ4, we investigated whether RecQ4[427–1116] is able to separate dsDNA by conducting fluorescence-based helicase assays. In addition, we generated a C-terminally extended RecQ4 variant, RecQ4[427-1208], to analyse the influence of the very C-terminal 92 amino acids (termed the CTD) on RecQ4s helicase activity.

Our *in vitro* analysis demonstrates, that RecQ4[427–1116] is able to separate a 3′-overhang (3′-OH) DNA-substrate upon ATP addition (Fig. 5a,b). Surprisingly, helicase activity is more than five-fold increased for the RecQ4[427–1208] variant. This implies that RecQ4[427–1116] features all basic structural requirements for dsDNA separation. However, the last 92 aa of RecQ4[427–1208] seem to contain additional elements, which are important for RecQ4s helicase activity. As a negative control, we analysed the two ATPase-deficient Walker A and Walker B variants (RecQ4[427–1208] K508A and D605A). As expected, no significant helicase activity was detected for both Walker-mutants (Fig. 5a,b), confirming that both, RecQ4[427–1116] and RecQ4[427–1208], are active helicases. To examine whether the different activities are a result of variations in DNA binding efficiency or ATPase rates, we performed equilibrium DNA titration measurements and NADH-coupled ATP-consumption assays. Both RecQ4 variants bind ssDNA with low nanomolar affinity (Fig. 5c). The DNA-binding data resulted in an apparent $K_d$ of $18.1 \pm 0.8$ nM for RecQ4[427–1116]. With an apparent $K_d$ of $4.6 \pm 0.4$ nM, RecQ4[427–1208] binds even tighter to ssDNA, indicating that the last 92 aa of RecQ4 contribute significantly to DNA binding. The ATP-dependent ATPase turnover data revealed equivalent $V_{max}$ values for both RecQ variants (Fig. 5d). These results indicate that the difference in helicase activity between the two

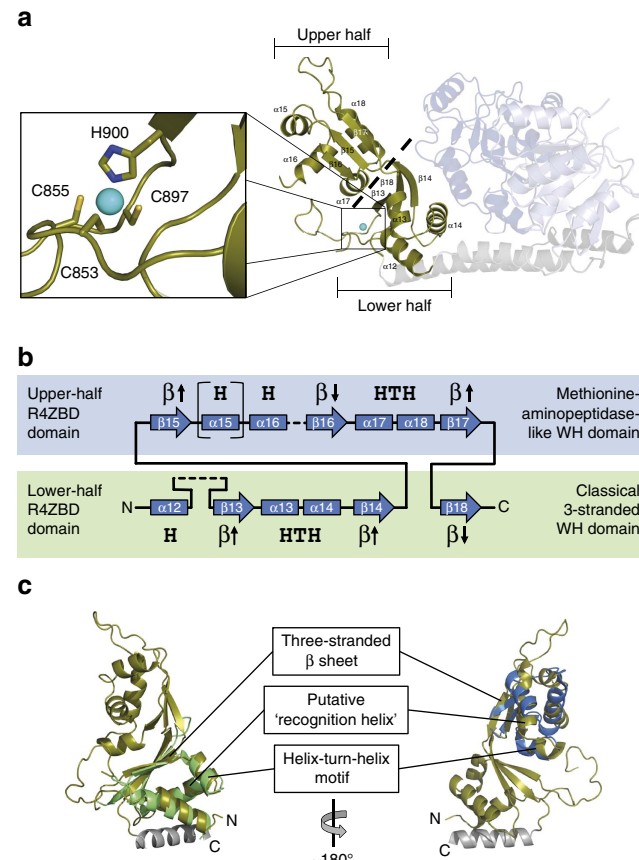

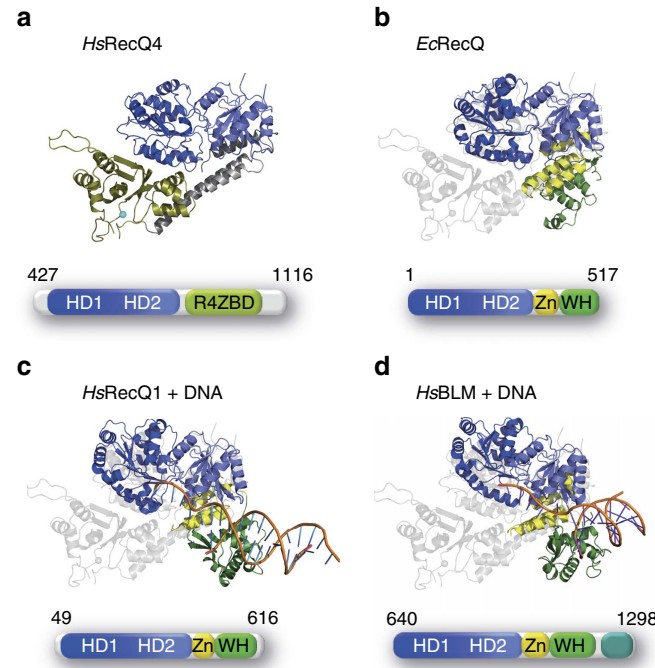

**Figure 2 | The R4ZBD features homology to WH domains.** (**a**) The R4ZBD can be separated into an upper and a lower half as indicated by the dashed line. Left: Enlarged view of the bound $Zn^{2+}$-ion and its coordinating residues, which are depicted in stick representation. (**b**) Illustration of R4ZBD secondary structure elements and their relation to the upper half (blue box) or the lower half (green box). Helices and β-strands are depicted as boxes and arrows, respectively, and are abbreviated with H and β. Black arrows indicate the orientation of the β-strand within its β-sheet. Regions, which are not resolved in the RecQ4 model, are depicted as dashed lines. Annotation of secondary structure elements is as in **a**. The upper-half WH domain represents an insertion into the lower-half WH domain between β-strands β14 and β18. Based on the order of secondary structure elements, the upper-half WH domain can be assigned to the group of methionine-aminopeptidase-like (MetAP-like) WH domains, while the lower-half WH domain features the characteristic sequence of the standard three-stranded WH-like group (Note: The R4ZBD-helix α15 is not part of the MetAP-like class of WH domains and is therefore depicted in brackets). (**c**) Superposition of WH domains identified by the DALI homology search with the two halves of the R4ZBD (shown in olive). The helix C-terminal to the R4ZBD, representing the proximal half of the bridging helix α19, was included in the structural homology search, yet was not present in any of the identified homologues. Left: WH domain of the bacterial transcription factor CtsR (green, PDB 3h0d). Domain orientation of the R4ZBD as in **a**. Right: WH domain of the human methionine aminopeptidase 2 (blue, PDB 5d6e). The R4ZBD is rotated by ∼180° with respect to the orientation in the left panel.

RecQ4 variants is a result of the increased DNA-binding affinity for RecQ4$^{427–1208}$. Interestingly, neither of the RecQ4 variants could efficiently separate DNA substrates with a dsDNA length of more than 18 base pairs. This is in agreement with previous reports and defines RecQ4 as a rather weak and none-processive helicase compared to other RecQ homologues[40,41].

**Figure 3 | Comparison of RecQ4 with other RecQ structures.** Structural alignment of the *Hs*RecQ4 structure (**a**) with the structures of *Ec*RecQ (PDB 1oyw) (**b**), *Hs*RecQ1 (PDB 2wwy) (**c**) and *Hs*BLM (PDB 4o3m) (**d**). Alignments were created in pymol. All structures were superimposed utilizing the isolated HD2 of *Hs*RecQ4. *Hs*RecQ4 is depicted in gray within each alignment for convenient comparison. Constructs used for crystallization of each RecQ variant are illustrated below the alignment, with the colour code and abbreviations as in Fig. 1a. The alignments demonstrate that the R4ZBD of RecQ4 assumes a unique structural fold, which does not resemble the typical RQC domain of other RecQ helicases. Furthermore, the R4ZBD acquires a unique position within the RecQ4 structure, adjacent to HD1, while all RQC domains are consistently located adjacent to the HD2. The two DNA-complex structures (**c,d**) depict a RecQ-conserved mode of DNA binding, with the ssDNA bound across the ATPase domain and the dsDNA portion bound via the WH domain. (Note: The *Hs*BLM structure (4o3m) also features the HRDC domain, which was omitted for the purpose of a uniform comparison).

**Functional analysis of the R4ZBD.** Due to the homology of the R4ZBD to prokaryotic DNA binding elements, we wanted to investigate whether the R4ZBD is able to emulate characteristic RQC domain functions, for example, DNA binding and helicase activity, despite their structural differences and positioning. To this end we generated a set of R4ZBD variants based on the RecQ4$^{427–1208}$ construct, and analysed them for DNA binding and helicase activity (Fig. 6a). To address the function of the upper WH domain, we generated a deletion variant, in which the entire upper WH domain (corresponding to aa 944–1032) was replaced by a GlyGly-linker. With respect to the lower WH domain, we targeted surface-exposed positively charged patches, which could be involved in DNA binding, by mutating corresponding Lys/Arg residues to Ala (R894A/R895A, K843A/R844A/R848A and K1048A). In case of the triple K843A/R844A/R848A variant, we additionally generated an inverse charge variant by replacing R844 with a glutamate residue (K843A/R844E/R848A). Fluorescence polarization analysis of all variants revealed no reduction in DNA affinity when compared to the RecQ4$^{427–1208}$ WT protein; with apparent $K_d$ values for all variants in the low nM range (Fig. 6b). The assessment of helicase activity also demonstrates that all variants are able to

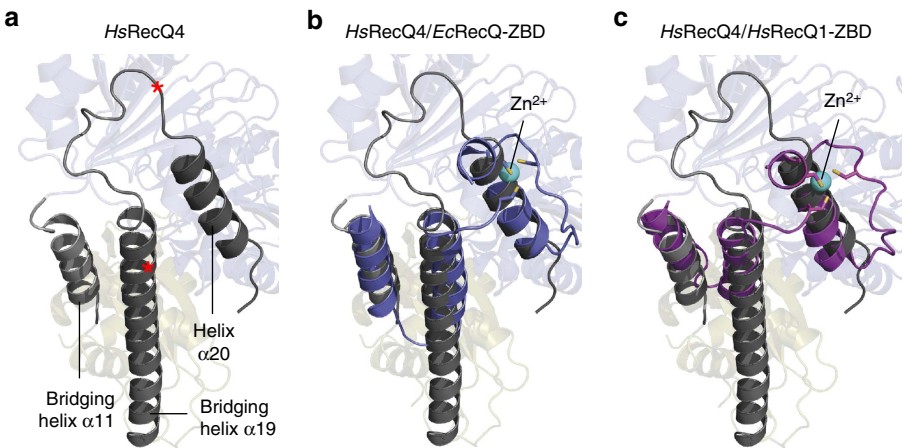

**Figure 4 | Three helices in RecQ4 mimic a RQC ZBD-like arrangement.** (**a**) The position of the two bridging helixes (α11 and α19) and helix α20 within the RecQ4 model are shown in dark grey. Red asterisks highlight the location of two early termination patient mutations (R1072X and Q1091X), which are associated with RAPADILINO syndrome (see Discussion). (**b**) Superposition of the three RecQ4 helices with the Zn-binding domain (ZBD) of *Ec*RecQ (depicted in blue). (**c**) Superposition of the three RecQ4 helices with the Zn-binding domain (ZBD) of *Hs*RecQ1 (depicted in magenta). (**b**,**c**) Bridging helix α11 and the downstream part of bridging helix α19 adopt the same position as the non-metal-coordinating helices, while helix α20 assumes the position of the in-line oriented $Zn^{2+}$-coordinating helices of the respective ZBDs. Alignments where extracted from the structural alignment as shown in Fig. 3.

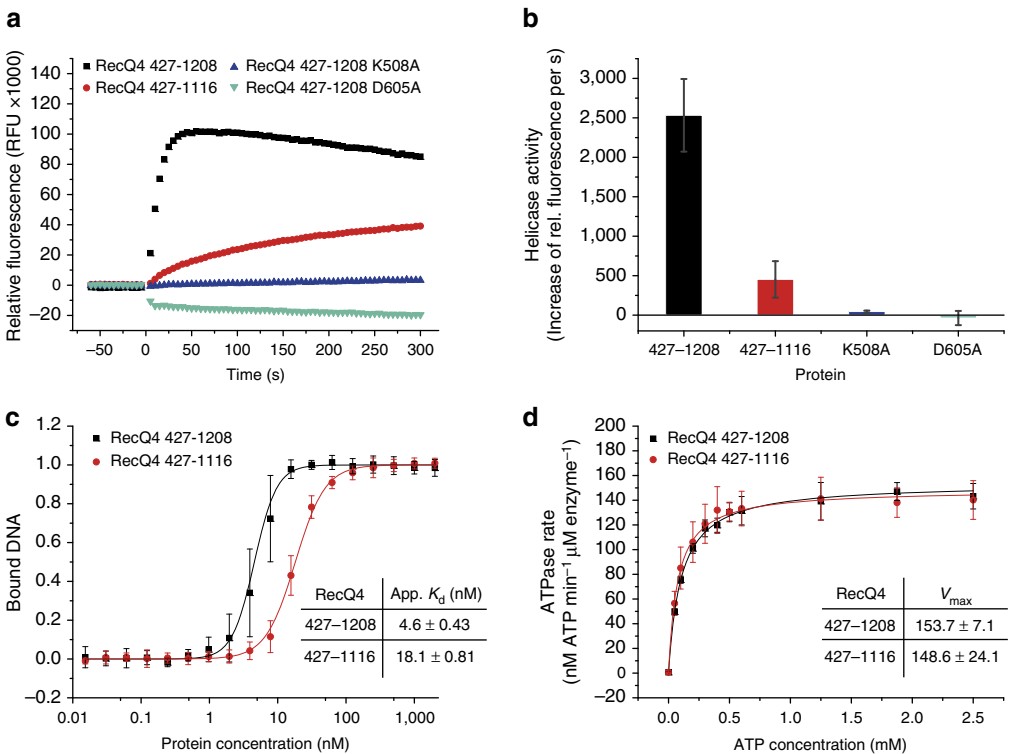

**Figure 5 | Biochemical analysis of RecQ4$^{427-1116}$ and RecQ4$^{427-1208}$.** (**a**) Analysis of RecQ4$^{427-1116}$ and RecQ4$^{427-1208}$ as well as the two ATPase deficient Walker-mutants (RecQ4$^{427-1208}$ K508A and D605A) by a fluorescence-based helicase assay. After ATP addition ($t_0$) the active helicases separate the quencher-labelled DNA strand from the Cy3-labelled DNA strand, detected by an increase in fluorescence. While both Walker-mutants display no helicase activity, RecQ4$^{427-1116}$ is an active helicase, although less efficient than RecQ4$^{427-1208}$. (**b**) Quantification of three individual helicase assays as in **a**. Bars represent the linear increase in relative fluorescence 10 s after ATP addition. Helicase activity is more than five-fold increased for the RecQ4$^{427-1208}$ variant. (**c**) Equilibrium DNA-binding data for a ssDNA substrate (T3-Cy3). DNA affinity is increased for the RecQ4$^{427-1208}$ variant, indicating that the CTD contributes to overall DNA binding. (**d**) Maximum ATPase rates ($V_{max}$) are equal for both RecQ4 variants, suggesting that the CTD does not influence overall ATP hydrolysis. Assays were performed at least three times using protein from two different purification batches. Error bars are defined as s.d.

separate a 3′-OH substrate (Fig. 6c). However, while the lower WH-point-mutation variants display a helicase activity indistinguishable from the WT, the Δ944–1032 deletion variant exhibits a reduction in the velocity of helicase activity corresponding to 53% of the WT protein (Fig. 6d).

**The role of the aromatic-rich loop.** RecQ/DNA complex structures display a conserved WH/dsDNA interaction (Fig. 3c,d), suggesting that the RQC domain and its positioning within the tertiary structure of RecQ helicases is crucial for helicase function. Since there is no RQC-like domain present

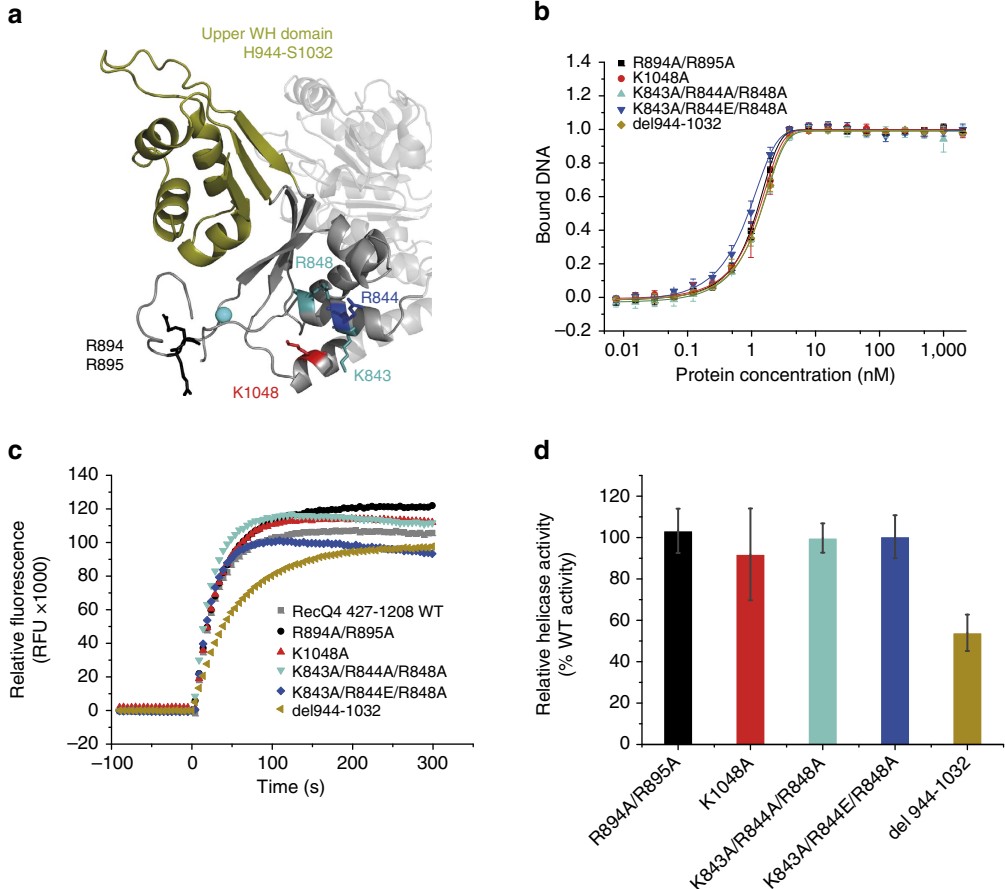

**Figure 6 | Biochemical analysis of the RecQ4[427-1208] R4ZBD variants.** (**a**) Overview of the location of all residues, analysed in the R4ZBD variants. The upper WH domain, depicted in olive, was replaced by a GlyGly-linker in the Δ944-1032 variant. (**b**) Equilibrium DNA-binding data for a ssDNA substrate (T3-Cy3). DNA-binding affinity is comparable to the RecQ4[427-1208] WT, with apparent $K_d$ values in the low nM range for all variants, indicating that the R4ZBD is not involved in ssDNA binding. (**c**) All analysed R4ZBD variants are able to separate a 3′-OH DNA substrate comparable to the WT protein. However, while all lower-WH-domain variants display a helicase activity, which is indistinguishable from the WT, the upper WH-domain deletion variant exhibits a reduced helicase velocity. (**d**) Quantification of three individual helicase assays as in **c**. Bars represent the linear increase in relative fluorescence 10 s after ATP addition. Depicted values are shown as relative to the WT protein. Assays were performed at least three times using protein from two different purification batches. Error bars are defined as s.d.

in RecQ4 and the R4ZBD does not adopt a RQC-typical position in the RecQ4 structure, it is questionable whether this RecQ-conserved mode of DNA-processing is present in RecQ4 as well. For bacterial RecQ helicases, Keck and coworkers described a conserved aromatic-rich sequence within the HD1, the ARL, which couples ssDNA binding to ATP hydrolysis[10,15]. The function of this sequence motif requires the trajectory of the ssDNA to enter the helicase from HD2 in the direction towards HD1, which positions the dsDNA in close proximity to the RQC-WH domain. These findings prompted us to investigate the significance of the ARL in RecQ4 and to evaluate whether this mode of DNA binding and DNA processing is conserved in RecQ4 as well.

The ARL in RecQ4 comprises the conserved residues W613, H615, F617 and R618 (Supplementary Fig. 1). To investigate their biochemical importance in RecQ4, we created single amino acid substitutions to leucines of all four conserved ARL-residues within the RecQ4[427-1208] variant. Equilibrium DNA-binding experiments confirmed that all ARL variants bind ssDNA with WT-like affinity (Fig. 7a). To explore the individual effects of each ARL substitution on the ability to hydrolyse ATP, we performed two variants of *in vitro* ATPase assays. First, we assessed the ATPase rates at saturating ssDNA conditions using different ATP

concentrations. As expected, none of the ARL variants could reach WT-like ATPase turnover rates (Fig. 7b). Compared to the RecQ4[427−1208] WT protein, the least affected ARL substitutions were W613L and H615L with 63% and 62% of WT-like ATPase activity, respectively. F617L and R618L were more strongly affected, with F617L being reduced to 35% and R618L to 21% of WT-like ATPase activity. The second assay variant illustrates the demand of ssDNA as a cofactor to perform ATP hydrolysis by varying ssDNA concentrations at saturating ATP conditions. The results demonstrate, that an increase in ssDNA concentration could not restore WT-like ATPase rates (Fig. 7c). The $K_{DNA}$ value represents the ssDNA concentration at half-maximal activity, indicating the demand of ssDNA required by the specific ARL variant to be activated. Except for F617L, which required less ssDNA compared to the WT protein, all other ARL variants were comparable to the WT (W613L and H615L) or needed significantly more ssDNA for activation (R618L) (Table 2). Notably, the F617L variant displayed a faint but measurable ATPase activity even in the absence of ssDNA, which was also observed for the phenylalanine substitution within the ARL of the *E. coli* RecQ helicase[10]. We then analysed the ability of the ARL variants to separate a 3′-OH DNA substrate in our fluorescence-based helicase assays (Fig. 7d,e). While the R618L helicase activity

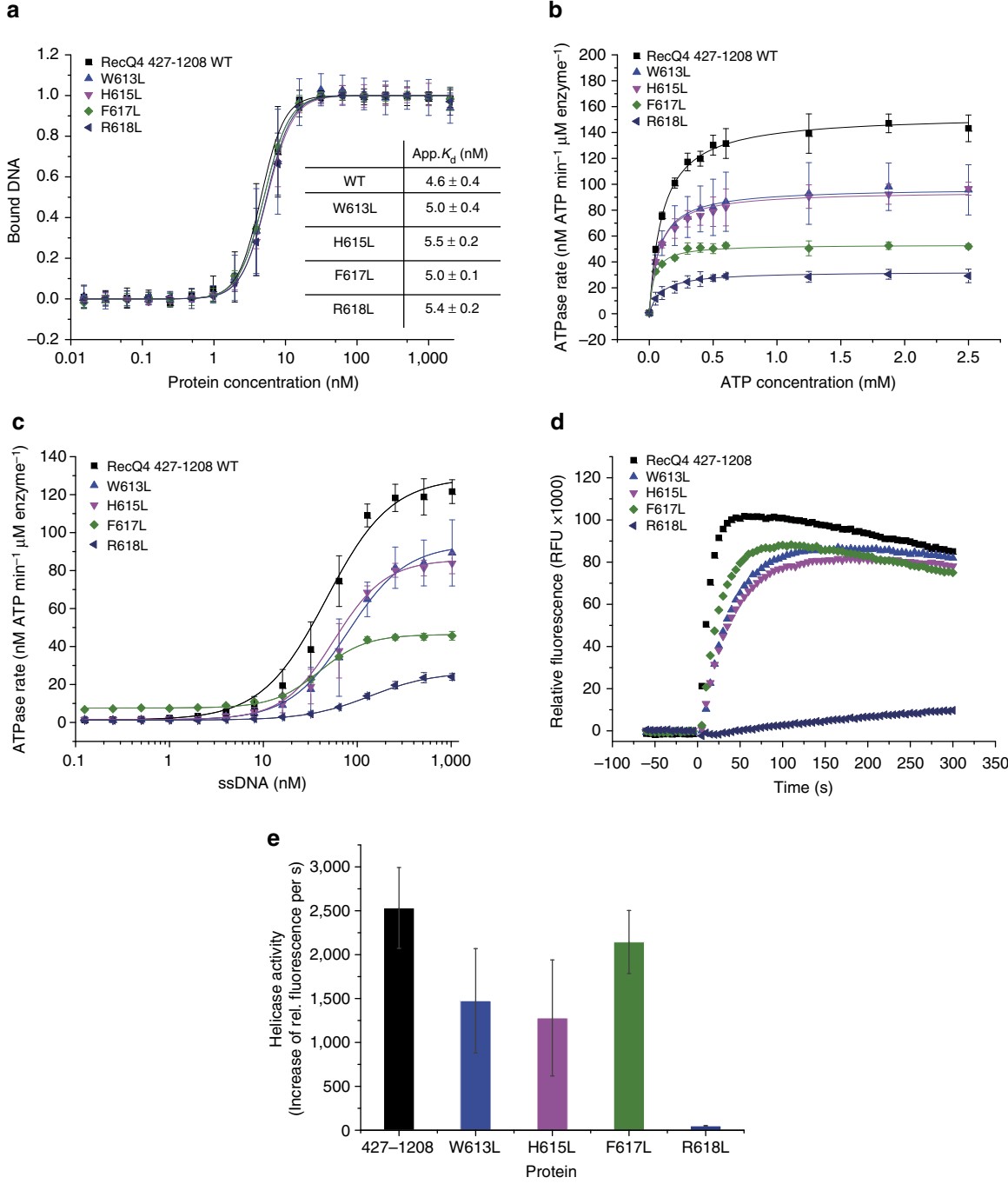

**Figure 7 | Biochemical analysis of the RecQ4[427-1208] ARL-variants.** (**a**) Equilibrium DNA-binding data for a ssDNA substrate (T3-Cy3). WT-like DNA binding is not affected by any single ARL substitution. (**b**) Each ARL substitution displays reduced ATPase activity compared to the WT protein. Least affected are W613L and H615L with approximately 60% of WT-like ATPase activity. F617L and R618L were reduced to 35% and 21% of WT-ATPase activity, respectively. (**c**) ssDNA-dependent ATPase rates of the ARL variants. W613L and H615L display a WT-like demand of ssDNA to hydrolyse ATP. While the demand for ssDNA is increased for R618L, it is slightly decreased in the F617L variant. Notably, F617L shows subtle ATPase activity even in the absence of ssDNA. (**d**) W613L, H615L and F617L display WT-like helicase activity, while the R618L variant is unable to separate the dsDNA substrate. (**e**) Quantification of three individual helicase assays as in **d**. Bars represent the increase in relative fluorescence 10 s after ATP addition ($t_0$). All ARL variants were tested towards their stability by a thermofluor ssay (Supplementary Fig. 4). Assays were performed at least three times using protein from two different purification batches. Error bars are defined as s.d.

is completely abolished, W613L, H615L and F617L showed only a slight reduction in helicase activity compared to WT RecQ[427−1208]. Interestingly, the helicase activity of F617L is not as much attenuated as observed for W613L and H617L. This mild reduction was unexpected, considering the strong impairment of the ATPase activity for the F617L variant.

## Discussion

In this study we present the crystal structure of the human RecQ4 helicase. Downstream of the HD2, we identified a unique domain, the R4ZBD. Reminiscent to other RecQ helicases, this domain features structural homology to WH domains and coordinates a $Zn^{2+}$-ion. Despite these similarities, however, the R4ZBD does

not share the tertiary structure of the conserved RQC-WH-fold, which is present in all described RecQ structures to date and which was reported to be indispensable for dsDNA strand separation in human RecQ1, WRN and BLM[11–14]. In addition to its unique fold, the R4ZBD assumes a special position adjacent to the HD1 of RecQ4 when compared to the position of the C-terminal domains of other RecQ helicases. A structural homology search of the isolated R4ZBD revealed pronounced homology to prokaryotic WH-like folds in the lower half of the domain and suggested the presence of a second WH domain in its upper half, albeit with less confidence. WH domains represent classical structural elements mediating protein-DNA interactions via a helix-turn-helix (HTH) core motif with a very broad functional versatility[42,43]. One way to achieve WH diversity is by variation of the order as well as by the addition or reduction of auxiliary secondary structure elements prior and after the HTH core[43]. These differences, in fact, classify the two WH domains discovered in the R4ZBD as well as the RQC-WH-fold as three different types of WH domains. While the upper R4ZBD-WH domain resembles a methionine-aminopeptidase-like WH domain, the lower R4ZBD-WH domain is categorized as a classical three-stranded WH-like domain[43] (Fig. 2b). A major difference between the R4ZBD- and the RQC-WH domains is the absence of the two-stranded β-hairpin element within the former, which destabilizes paired bases at the ssDNA/dsDNA junction and separates dsDNA by utilizing the translocation force along the DNA exerted by the ATPase domain. Our mutational and functional analysis of the R4ZBD further demonstrates that neither of the R4ZBD WH motifs is required for DNA binding or dsDNA strand separation of a 3′-OH DNA substrate (Fig. 6b-d). In addition, the putative recognition helix of the HTH motif within the lower WH domain is rather special since it harbours four glutamic acid residues, two of which are mediating surface contacts to the neighboring HD1, arguing against DNA binding via this WH/HTH motif. Combined, our data suggest that the R4ZBD is not directly involved in dsDNA strand separation but could pose as a specific DNA interaction module for more complex DNA-metabolism intermediate structures. In addition to protein-DNA interactions, WH domains are known to act as interface for protein–protein interactions[43,44,45], suggesting a possible role for the R4ZBD-WH domains as a platform to interact with other proteins. A prominent interaction partner of RecQ4 is poly(ADP-ribose) polymerase-1 (PARP-1), whose interaction was mapped to the C terminus of RecQ4 (aa 833–1,208)[46], a region which includes the R4ZBD.

Our structural alignment between RecQ4 and other known RecQ structures revealed that RecQ4s bridging helix α11, the downstream part of bridging helix α19 and helix α20, together, mimic the conserved structure and exact position of the RQC-$Zn^{2+}$-binding domain of other RecQ helicases (Fig. 4b,c), with the notable difference that RecQ4 does not coordinate a zinc ion at this position. We propose that the RQC-$Zn^{2+}$-binding domain and its RecQ4 counterpart are primarily relevant for protein stability. This hypothesis is supported by two reported patient mutations within RecQ4, Arg1072X and Gln1091X[47], which eliminate helix α20 and all following amino acids due to a premature stop-codon (Fig. 4a). Each of the two mutations was reported heterozygous with an Ala420-Ala463del deletion mutation within the second allele and both patients were diagnosed with RAPADILINO syndrome[8]. To characterize the effects of the premature termination biochemically, we generated a RecQ4[427–1090] variant, which represents the Glue1091X patient mutation. In support of our hypothesis, we could not achieve soluble expression of this variant despite the fact that the only difference to the soluble RecQ4[427–1116] variant is the absence of

helix α20. The Ala420-Ala463del mutation was reported to be a founder mutation causing the RAPADILINO syndrome, as there are many homozygous patients reported, especially in the Finnish population[47]. However, with the second allele of the RecQ4 enzyme being compromised in its stability by the Arg1072X or Glue1091X early termination mutation, the defect of the Ala420-Ala463del mutation cannot be compensated, leading to the disease phenotype. Unfortunately, our model provides no indication of the structural relevance of the Ala420-Ala463del mutation, as most of the deleted residues are not resolved in the structure. The deletion does not include any secondary structure elements of the HD1, yet affects residues of the major NTS within the N-terminus of RecQ4 (aa 363–492)[26], suggesting a compromised subcellular localization of the RecQ4_Ala420-Ala464del variant rather than impaired helicase activity.

Our in vitro biochemical analysis of the RecQ4[427–1116] variant provides evidence that our model features all necessary structural elements for ATP- and DNA-dependent helicase activity. Interestingly, this activity is substantially increased when the RecQ variant features the CTD (aa 1117–1208). While both protein variants, RecQ4[427–1116] and RecQ4[427–1208], display essentially the same ssDNA-dependent ATPase rates, DNA binding is about fourfold increased by the presence of the CTD, indicating that the CTD is significantly involved in overall DNA-binding. Consequently, increased DNA-binding mediated by the CTD might be responsible for the substantial difference in helicase activity between both RecQ4 variants. Since the absence of the CTD does not abolish helicase activity, we hypothesize that RecQ4's helicase mechanism might not be solely based on a physical wedge element, as described for the RecQ1, WRN and BLM helicases[11–14]. It is tempting to speculate that the helicase function in RecQ4 is thus achieved by a specific arrangement of the DNA substrate within the protein–DNA complex, similar to the dsDNA separation mechanism proposed for bacterial RecQs[15]. In analogy to the bacterial RQC domain, the CTD could bind and arrange the dsDNA in a special conformation, forcing a critical angle between dsDNA and ssDNA at the ssDNA/dsDNA junction, resulting in the destabilization of the paired bases and thereby permitting efficient dsDNA unwinding via translocation. However, both strand separation mechanisms are not mutually exclusive and in the absence of a DNA-bound crystal structure of RecQ4 we cannot exclude the presence of a structural element, which could serve as a dsDNA strand separation wedge.

To assess the potential trajectory of the ssDNA in RecQ4, we investigated whether the mode of ssDNA binding and translocation is conserved between RecQ4 and the EcRecQ helicase and performed mutational studies on the ARL. Our analysis demonstrates, that the ARL in RecQ4 acts in a similar manner as in EcRecQ, recognizing ssDNA binding and thereby enabling ATP hydrolysis. Similarly to EcRecQ, individual ARL mutations did not alter ssDNA-binding efficiency compared to the WT protein, indicating that the reduction in activity is based on the disrupted ssDNA sensor function. We identified W613 and H615 to act predominantly in this ssDNA sensor function, as the ATPase rates of their respective leucine mutations were affected to the same degree as their helicase activity (Table 2). F617L and R618L displayed the strongest reduction in their ATPase rates, yet in contrast to W613L and H615L, both variants exhibit a disproportional reduction in their helicase activity. These findings suggest that F617 and R618 interact directly with the ssDNA strand and might thus be implicated in ssDNA translocation, while W613 and H615 mediate secondary contacts to residues of the helicase domains on ssDNA binding by the ARL and thereby conduct the signal of DNA binding to enable ATP hydrolysis. The F617L variant displayed almost WT-like helicase

**Table 2 | Summary of the RecQ4 ARL analysis presented in Fig. 7a–e.**

|  | WT | W613L | H615L | F617L | R618L |
|---|---|---|---|---|---|
| % WT ATPase | 100 | 63.2 | 61.7 | 34.7 | 21.1 |
| % WT Helicase | 100 | 58.2 | 50.5 | 84.7 | 2.0 |
| $K_{DNA}$ (nM) | 46.9 ± 7.3 | 79 ± 9.6 | 57.1 ± 4.3 | 39.7 ± 2.1 | 137.1 ± 10.9 |
| Efficiency (Helicase/ATPase) | 1.00 | 0.92 | 0.82 | 2.44 | 0.1 |

ARL, aromatic-rich loop; WT, wild type.
Efficiency is defined by the amount of helicase activity (% WT Helicase) per ATPase activity (% WT ATPase) for each ARL variant. Therefore, efficiency values of 1 indicate a direct correlation between helicase- and ATPase reduction. F617L and R618L display a disproportional reduction in ATPase and helicase activity, resulting in increased efficiency for F617L and reduced efficiency for R618L, respectively.

activity while being severely reduced in its ATP turnover rates. This argues for a decelerating effect of F617 on ssDNA translocation, which is abolished by the leucine mutation. In contrast, the R618L substitution completely eliminates helicase activity while still exhibiting ssDNA-dependent ATP hydrolysis suggesting a propulsive role for R618 in ssDNA translocation. The leucine substitution eliminates this active role of R618, thereby rendering the helicase unable to use the energy of hydrolysed ATP for protein translocation and consequently, helicase activity. Our mutational analysis of the RecQ4 ARL thus demonstrates that the binding mode for ssDNA across the surface of the ATPase domain, in close proximity to the ARL, is conserved between RecQ4 and *Ec*RecQ and should thus also resemble the DNA substrate binding as observed for eukaryotic RecQ helicases (Fig. 3c,d). Hence, we can assume that the ssDNA enters the RecQ4 helicase via HD2 and is directed towards HD1, positioning the dsDNA entity in close proximity to the RecQ4 CTD. In agreement with our ARL data, the RecQ4 structure exhibits a highly electropositive cleft between the R4ZBD and HD1, poised to bind the negatively charged ssDNA backbone (Supplementary Fig. 5). However, further studies are required to establish a comprehensive model for DNA-binding and translocation by RecQ4.

During the preparation of this manuscript a study by Mojumdar *et al.*[48] was published, which suggested that RecQ4 features a functional RQC domain that is essential for RecQ4 activity. This conclusion was in part based on the biochemical analysis of various cysteine mutations, which were proposed to be responsible for the formation of two individual zinc-binding sites. While our RecQ4[427–1116] structure demonstrates that RecQ4 does not feature a canonical RQC domain, our data confirms one zinc-binding site, which is located in the R4ZBD. However, the remainder of the analysed cysteine residues do not participate in any further zinc coordination in our structure. The authors also presented low-resolution SAXS data for a RecQ4[445–1112] protein construct. Unfortunately, as the R4ZBD and the RQC domain are similar in size and adopt parallel but opposing positions with respect to the ATPase domain, the authors were misled to the conclusion that RecQ4 would feature a RQC domain.

In summary, our analysis of the RecQ4 protein provides valuable insights into the tertiary structure of the human RecQ4 helicase as well as its unique C-terminus and demonstrates the extensive structural differences between RecQ4 and other RecQ helicases. Future studies are required to address the importance of the two WH domains within RecQ4s R4ZBD and dissect their involvement in RecQ4 function and RecQ4-associated diseases. Unexpectedly, the RecQ-conserved $Zn^{2+}$-binding domain is mimicked in location as well as structure in RecQ4 by a set of three helices; an arrangement we propose to have a predominant function towards protein stability. This is supported by an analysis of three reported RecQ4 patient mutations, leading to the RAPADILINO syndrome. Finally, the RecQ4 structure presented here highlights the structural and functional diversity between

RecQ4 and other human RecQ helicases. Thus RecQ4 has the potential to become a promising therapeutic target to approach RecQ4 upregulation in prostate, cervical and breast cancers.

## Methods

**Cloning.** The cDNA sequences corresponding to RecQ4 (UniProtKB O94761) variants 427–1116 and 427–1208 were cloned into the pETM-22 vector (EMBL, Heidelberg, Germany) between the 3C recognition sequence and BamH1 restriction site using the sequence-and-ligation-independent-cloning (SLIC) method[49]. All constructs featured an N-terminal thioredoxinA solubility tag, followed by a Hexahistidine tag and the 3C protease cleavage site. Single amino acid substitutions were generated according to the QuickChange site-directed mutagenesis protocol (Stratagene). A list of all primer sequences used for cloning and mutagenesis can be found in Supplementary Table 3.

**Protein expression and purification.** All RecQ4 variants were expressed in Terrific Broth media utilizing the BL21Star (DE3) *E. coli* strain (Life Technologies), which was supplemented with the pRARE2 plasmid from the Rosetta2 *E. coli* strain (Novagen). Cells were grown at 37 °C to an OD of 2.5 and transferred to 18 °C. Recombinant protein expression was induced by the addition of 0.1 mM IPTG. After 18 h, cells were collected and resuspended in 10 volumes of lysis buffer (50 mM HEPES pH 8.0, 50 mM NaCl, 20% Glycerol, 2.5 mM $MgCl_2$, 1 mM TCEP), complemented with cOmplete EDTA-free protease inhibitor tablets (Roche), 1 mg ml$^{-1}$ lysozyme and DNAseI. After cell disruption the crude extract was centrifuged at 4 °C for 1 h at 38,000 g. The supernatant was loaded on a pre-equilibrated 100 ml Heparin FF sepharose column (Heparin buffer: 20 mM HEPES pH 8.0, 10 mM NaCl, 10% Glycerol, 2.5 mM $MgCl_2$, 1 mM TCEP). The column was washed with 1.1 CV (column volume) Heparin buffer supplemented with 30 mM NaCl and the protein was eluted with 3.5 CV Heparin buffer supplemented with 1 M NaCl. The eluate was collected and directly applied to a 5 ml HisTrap FF crude column, pre-equilibrated with IMAC buffer (20 mM HEPES pH 8.0, 300 mM NaCl, 10% Glycerol, 15 mM imidazole, 2.5 mM $MgCl_2$, 1 mM TCEP). The HisTrap column was washed with 3 CV of IMAC buffer supplemented with 25 mM imidazole and the protein was eluted using a linear gradient over 5 CV up to a concentration of 200 mM imidazole. Appropriate fractions were pooled and the Trx-His(6x)-tag was removed by incubation with 3C protease. After tag removal, the HisTrap eluate was concentrated and applied to a Superdex 200 16/600 size exclusion column using Gel-filtration buffer (20 mM HEPES pH 8.0, 200 mM NaCl, 10% Glycerol, 2.5 mM $MgCl_2$, 1 mM TCEP). Appropriate fractions were pooled and the salt concentration was reduced to 35 mM NaCl. The solution was applied to a MonoS 5/50 GL column, pre-equilibrated with MonoS buffer (20 mM HEPES pH 7.2, 10 mM NaCl, 10% Glycerol, 2.5 mM $MgCl_2$, 1 mM TCEP). The column was washed with 5 CV of MonoS buffer and the protein was eluted with a linear gradient over 40 CV to a concentration of 200 mM NaCl. Appropriate fractions were pooled and concentrated. As a final polishing step, all RecQ variants were applied to an analytical Superdex 200 Increase 10/300GL size exclusion column, using protein-storage buffer (10 mM HEPES pH 8.0, 100 mM NaCl, 10% Glycerol, 2.5 mM $MgCl_2$, 1 mM TCEP). Appropriate fractions were pooled and concentrated. Protein was flash frozen in liquid nitrogen and stored at − 80 °C.

**Crystallization and structure determination.** Crystals were obtained with the hanging-drop vapour diffusion method using 2.8–3.0 M sodium formate and 0.1 M imidazole pH 9.0 as precipitant and a protein concentration of 3 mg ml$^{-1}$ with a protein-to-precipitant ratio of 2:1. Crystallization was carried out at 20 °C. Crystals were flash-frozen in mother liquor supplemented with 25% (v/v) glycerol. Diffraction data were collected at a temperature of 100 K at beamlines ID29 (ESRF, Grenoble, France) and BL14.1 (BESSY, Berlin, Germany). Data were integrated and scaled with XDS[50] and Aimless[51]. Trials to solve the structure by molecular replacement were unsuccessful. As diffraction data indicated the presence of an anomalous scatterer, an X-ray fluorescence scan was performed. This indicated the presence of Zn in the crystal. The Crank2 pipeline[52] was used to solve the structure in space group P3$_1$21 by the single-wavelength anomalous

### Table 3 | Oligonucleotide sequences for the preparation of DNA substrates.

| Name | Label | Sequence (5′ to 3′) |
|---|---|---|
| T3-Cy3 | 5′ Cyanine-3 | CCATTCCACCCTCTATTTTTTTTTTTTTTTT |
| B1-Dab | 3′ Dabcyl | TAGAGGGTGGAATGG |
| T30 | — | TTTTTTTTTTTTTTTTTTTTTTTTTTTTTT |

dispersion (SAD) method with data collected at the Zn-peak wavelength (Table 1). A single $Zn^{2+}$-ion was located with AFRO[53]/CRUNCH2[54]. Several cycles of phasing, density modification and chain-tracing with REFMAC[55], PARROT[56] and Buccaneer[57] resulted in an initial model consisting of 581 residues in 17 fragments. This model was manually completed with COOT[58]. Restrained refinement of coordinates, B-factors and TLS-parameters was performed with Phenix[59] against data extending to a resolution of 2.75 Å. The final model was refined to an $R_{work}$ of 18.3% and an $R_{free}$ of 23.1% with 97% of the residues in the most favoured regions of the Ramachandran plot and none within disallowed regions.

**DNA substrate preparation.** Synthesized DNA oligonucleotides (biomers.net) were resuspended in sterile milliQ water to a concentration of 200 μM and stored at −20 °C. Oligonucleotide sequences used for DNA substrate preparation are listed in Table 3. DNA annealing was performed in 20 mM HEPES pH 8.0 and 100 mM NaCl. 100 μl reactions including 10 μM of each, the Top and Bottom strand, were heated to 95 °C for 5 min and gradually cooled to 15 °C at a rate of 1 °C min$^{-1}$ using a PCR thermal cycler (Eppendorf). Annealed DNA substrates were kept at 4 °C in dark storage.

***In vitro* helicase assays.** Helicase assays were performed in a Clariostar microplate reader (BMG LABTECH) at 25 °C in 50 μl reactions containing 1 μM protein and 50 nM of a 15 nt-3′overhang (OH) DNA substrate in assay buffer (20 mM HEPES pH 8.0, 10 mM NaCl, 5% Glycerol, 1 mM $MgCl_2$, 0.5 mM TCEP). The DNA substrate (T3-Cy3 annealed to B1-Dab) contained a 3′-15 nt polyT ssDNA loading site and a 15 nt dsDNA part with a generic sequence. After recording baseline fluorescence for 60 s, the reaction was initiated by adding ATP to a final concentration of 1.25 mM. The helicase reaction, visualized by increasing Cy3-fluorescence as the quencher-labelled bottom-DNA strand is separated from the Cy3-labelled top-DNA strand, was recorded by monitoring the Cy3-fluorescence (Excitation 530 nm / Emission 580 nm) for 5 min. Measurements using $H_2O$ in place of ATP as well as reactions with buffer instead of protein served as blank and were subtracted from the ATP-data. Helicase rates were calculated based on the linear increase in fluorescence for the first 10 s after ATP addition. All reactions were carried out in triplicates using at least two different protein batches.

***In vitro* ATPase assays.** ATPase turnover rates were determined using an NADH-coupled ATP consumption assay, where ATP hydrolysis, performed by the helicase, is coupled to the proportional decrease of NADH concentration through an ATP regeneration system based on phosphoenolpyruvate (PEP), pyruvate kinase (PK) and lactate dehydrogenase (LDH). NADH absorbance at 340 nm was recorded over 30 min and converted into ATP concentration using the molar extinction coefficient of NADH. All reactions were carried out at 30 °C using the Clariostar microplate reader (BMG LABTECH). Two types of ATPase assays were performed: (1) ATP-concentration dependent ATPase rates were measured in 50 μl reactions containing 0.3 mM NADH, 2.7 U LDH, 1.8 U PK, 2 mM PEP and 1 μM T30-ssDNA substrate in assay buffer (20 mM HEPES pH 8.0, 10 mM NaCl, 5% Glycerol, 1 mM $MgCl_2$, 0.5 mM TCEP). The reactions were supplemented with 0.3 μM RecQ4 helicase variant and initiated by addition of ATP with concentrations ranging from 0 mM to 2.5 mM. The ATP-concentration dependent ATPase data were fitted to the Michaelis-Menten-equation to calculate the maximal ATPase rates ($V_{max}$). (2) ssDNA-dependent ATPase rates were conducted in the same way as in (1) with the exception that the reactions varied in their T30-ssDNA concentration, representing a serial dilution from 1.024 μM to 0.125 nM, and a standard ATP concentration of 1.25 mM was used to initiate the reaction. ssDNA-dependent ATPase data were plotted to a logarithmic ssDNA-concentration scale and fitted to a logistic fitting function in OriginPro (OriginLab). $K_{DNA}$ values represent the ssDNA concentration required for half-maximal activity. All experiments were performed in triplicates with protein from at least two different batches.

**Fluorescence polarization assays.** Fluorescence polarization measurements were performed in 100 μl reaction volume containing assay buffer (20 mM HEPES pH 8.0, 10 mM NaCl, 5% glycerol, 1 mM $MgCl_2$, 0.5 mM TCEP) and 0.5 nM Cy3-labelled ssDNA substrate (T3-Cy3). Each reaction was supplemented with RecQ4 protein, representing a serial dilution from 2 μM to 3.8 pM. Fluorescence polarization values for each RecQ4 variant concentration were determined by measuring the parallel and perpendicular fluorescence of the Cy3-labelled ssDNA

(Excitation 530 nm/Emission 580 nm) using the Clariostar microplate reader (BMG LABTECH). Maximum and minimum polarization values indicate fully bound and unbound ssDNA, respectively, and were used to normalize the ssDNA binding curves. Normalized values were plotted to a logarithmical scale of ssDNA concentration and fitted using the logistic fitting algorithm in OriginPro (OriginLab). The ssDNA concentration at half-maximal binding represents the apparent binding constant ($K_d$) for each RecQ4 variant.

**Thermofluor assay.** To confirm intact folding of all RecQ4 variants, we performed a Thermofluor analysis, in which a slow and controlled heat increase leads to the thermal denaturing of the tested RecQ4 variants. This is visualized by an increase in the fluorescence signal, arising from a dye (SYPRO Orange), which binds to hydrophobic/unfolded patches and fluoresces. Proteins with a similar structural fold therefore show overlapping thermal denaturing curves. For each RecQ4 variant, 5 μl of protein (1.74 mg ml$^{-1}$) were mixed with 1 μl of a 2.5% SYPRO Orange solution and 19 μl storage buffer (10 mM HEPES pH 8.0, 100 mM NaCl, 10% glycerol, 2.5 mM $MgCl_2$, 1 mM TCEP) and heated gradually to 95 °C with a step size of 1 °C min$^{-1}$ using a qPCR machine (Stratagene Mx3005P), which permits simultaneous SYPRO Orange fluorescence monitoring (Excitation 492 nm/Emission 610 nm). Data were analysed using Excel (Microsoft) and OriginPro (OriginLab).

**Data availability.** Coordinates and structure factors have been deposited in the Protein Data Bank (PDB) with the accession code 5LST. The data that support the findings of this study are available from the corresponding author upon reasonable request.

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

## Acknowledgements

We thank Vilhelm Bohr and Deborah Croteau (NIA, Baltimore (MD), USA) for providing the plasmid with the human RecQ4 DNA sequence as well as for helpful discussions. We thank the staff at the synchrotron beamlines ID29 (ESRF, Grenoble, France) and BL14.1 (BESSY II, Berlin, Germany) for technical support. Furthermore, we thank the members of our structural biology group Jochen Kuper for careful reading of the manuscript and Wolfgang Koelmel for collecting data sets, which were used in this study. This study was supported by a grant of the German Excellence Initiative to the Graduate School of Life Sciences, University of Wuerzburg.

## Author contributions

S.K., F.S. and C.K. conceived and designed the experiments. S.K. performed cloning, protein expression, protein purification, crystallization and conducted the experiments. S.K. and F.S. collected crystallographic data. F.S. analysed the X-ray diffraction data and performed model building and structure refinement. S.K. wrote the manuscript.

## Additional information

**Competing interests:** The authors declare no competing financial interests.

