## [Peer review file · Nature Communications]

Reviewers' comments:

Reviewer #1 (Remarks to the Author):

Kaiser et al. present the structure of the human RecQ4 helicase; a thus-far poorly structurally characterised member of the RecQ family. They identify a novel hybrid zinc-binding/WH domain which differs from classical eukaryotic RecQ members and suggest that the unwinding mechanism more closely resembles that of the bacterial proteins. The technical quality of the experiments reported appears good, with crystallographic and biochemical data of high quality that support the principle claims of the manuscripts. The figures are clear and informative but there are some areas in which the text could be improved (detailed below). Overall, I think this is a nice paper. The determination of the structure is an important advance for the the field. However, the proposed strand separation mechanism, while plausible is at the stage speculative and may require further support.

Major points:

1. The authors identify the R4ZBD domain, consisting of a "lower" winged-helix (WH) domain with a degenerate WH-fold insertion. This lower WH domain is a structural homolog of dsDNA-binding bacterial transcription factors. Does the WH domain in the R4ZBD domain bind DNA? By comparison to known dsDNA-binding WH domains, can DNA interacting residues (e.g. lysine etc.) be identified? If so, these should be mutated and the DNA-binding properties of the intact protein re-determined. Alternatively, the isolated R4ZBD domain could be made recombinantly and tested for DNA-binding activity.
2. It is possible to delete either the entire R4ZBD domain or upper WH domain alone (for example be replacing the domain with a short flexible linker sequence)? If it is indeed a protein-protein interaction module as suggested it may be dispensable for the helicase activity of the protein.
3. By analogy to RQC domains of other RecQ proteins that remain in the same position regardless of the presence or absence of DNA, the authors claim that the observed position of the R4ZBD in hsRecQ4 is not simply due to lack of DNA substrate (lines 185-188). I don't really think is a strong enough argument - after all, they are structurally different domains that likely have differing functions. Also, large domain motions have been observed in other DNA helicases upon substrate binding. Testing this (perhaps by SAXS or FRET studies) would substantially enhance the study. Alternatively, this claim should be dropped.
4. The authors carry out a helicase activity assay of a C-truncated construct (427-1116) as well as one including the C-terminus (427-1208) and show that longer version as about 5-fold more active than the truncation (figures 5a and 5b). They conclude that the shorter construct can support helicase activity (lines 220-221). It would be interesting if the authors could include in their assay a known helicase-defective point mutant (for example in one of the signature helicase motifs) to allow better assessment of the absolute activity of the shorter protein.

Minor Points:

1. In the introduction and abstract it is stated that RecQ4 lacks characteristic RecQ domains. How then is it defined as a member of the RecQ family?
2. Mutations in RecQ4 are linked to several diseases (introduction, lines 58-59). It would be useful if the authors could summarise the phenotypes of these syndromes, so that the reader might better appreciate the physiological effects of RecQ impairment.

3. It is not really clear what is meant by the use of the word "topologically" (line 138). This is also confusing as the authors later claim that the R4ZBD domain cannot be separated into two units (lines 144-145).

4. How do the authors know that the R4ZBD is packed "loosely" against HD1 (lines 152-153). Presumably if there are salt bridges and hydrogen bonds between the domains, they are not loosely bound!

5. Line 199. none-Zn should be non-Zn I think.

6. Labelling locations of of the N- and C-termini as well as sequence numbers at the positions of the domain boundaries in figure 1b would aid reader comprehension.

Reviewer #2 (Remarks to the Author):

The paper "Human RecQ4 structure and function reveals bacterial ancestry; a potential target in cancer therapy" by Kaiser et al presents the crystal structure of human RecQ4 helicase ATPase core domain with a unique C-terminal extension domain as well as biochemical characterization of several mutants. Experimental data are solid and the structure of unique R4ZBD domain is very interesting. Unfortunately, the properties and function of novel domain were not addressed experimentally, main conclusions are very speculative and, as a result, the paper present only incremental advance in the field. Therefore, it is not of interest for a wide research community and has to be published in more specialized journal.

Specific concerns:

1) The authors did not study functional property of novel structural domain, which significantly deviates from RQC domain mentioned by the authors as a part of functional core in all other RecQ helicases. Instead they targeted 1) the C-terminal extension with unknown structure, and 2) ARL motif already studied in other helicase. The former resulted in additional role of C-terminal domain for DNA binding, although whether it is due to additional DNA-binding region or structural role for core stability is unknown. The later just confirmed identical function of ARL motif from other helicase. Thus, although experiments were properly conducted, results are not significant. It is unclear why the authors dedicated majority of the paper to structural comparison of the novel R4ZBD domain with known structures, but did not attempt to test its functionality biochemically.

2) The choice of the title is odd and some conclusions are highly speculative and not supported by experimental data.

First, bacterial heritage does not warrant drug design advantage to target specific protein. Any structural differences can be used for inhibition specificity. Moreover, the paper only briefly refers to rationale of RecQ4 targeting for drug design without addressing it in more details. This is not a topic of current research and should not be in a title of the paper.

Second, the conclusion about protein similarities are based on the absence of the b-hairpin in RecQ4 combined with similarities in ARL motif leading to proposal that RecQ4 should bend DNA during unwinding to a larger degree than other human RecQs and similarly to that of the bacterial RecQ (what is the difference in bending degree is not discussed). This logic is odd at multiple levels. Only small portion of novel domain overlaps spatially with RQC and large structural differences point to differences in DNA binding between RecQ4 and remaining RecQs with the conserved RQC domain, rather than similarities. Note, that bacterial homolog does contain b-hairpin, even though its mutation does not affect unwinding to the same degree as in other helicases.

It is also unclear why similar function of RecQ4 ARL motif with the one in bacterial RecQ distinguish it from other human RecQs? Are there known differences in properties of ARL motifs?

Minor points:

- 1) It will be helpful to have one structural picture with numbers of important residues in main text or at least the sequence with secondary structure elements and numbering. It is difficult for the reader to follow description and discussion otherwise.
- 2) Spell HD1,2 abbreviation at the beginning.
- 3) Lane 113: "rore"
- 4) The significance of similar location of helices with zinc finger of RQC is unclear. This region does not form contact dsDNA and is a linker between HD2 and RQC, location of which is very different from that of R4ZBD. If R4ZBD binds DNA, then it should either swivel around α -helices or support very different conformation of DNA.
- 5) Discussion is extremely lengthy, repetitive and excessively speculative, even for structural paper.

Reviewer #1

Kaiser et al. present the structure of the human RecQ4 helicase; a thus-far poorly structurally characterized member of the RecQ family. They identify a novel hybrid zinc-binding/WH domain which differs from classical eukaryotic RecQ members and suggest that the unwinding mechanism more closely resembles that of the bacterial proteins. The technical quality of the experiments reported appears good, with crystallographic and biochemical data of high quality that support the principle claims of the manuscripts. The figures are clear and informative but there are some areas in which the text could be improved (detailed below). Overall, I think this is a nice paper. The determination of the structure is an important advance for the field. However, the proposed strand separation mechanism, while plausible is at the stage speculative

and may require further support.

Major points:

1. The authors identify the R4ZBD domain, consisting of a “lower” winged-helix (WH) domain with a degenerate WH-fold insertion. This lower WH domain is a structural homolog of dsDNA-binding bacterial transcription factors. Does the WH domain in the R4ZBD domain bind DNA? By comparison to known dsDNA-binding WH domains, can DNA interacting residues (e.g. lysine etc.) be identified? If so, these should be mutated and the DNA-binding properties of the intact protein re-determined. Alternatively, the isolated R4ZBD domain could be made recombinantly and tested for DNA-binding activity.

The recognition-helix of the HTH motif within lower-WH-domain appears to be rather special as it features four negatively charged glutamate residues, two of which are involved in surface contacts to the neighboring HD1. Based on this observation, we consider it unlikely to be involved in a canonical DNA substrate recognition. However, the area adjacent to the lower WH-HTH motif harbors several positively charged and surface exposed residues, which could potentially be involved in DNA binding (illustrated in Fig. 6a and corresponding to the electropositive patch #2 in supplementary Fig. 5). To address this issue, we initially approached recombinant expression of the isolated R4ZBD and although we achieved soluble expression of the protein domain, yields were very low and subsequent purification proved to be problematic. As an alternative, we created variants of the RecQ4⁴²⁷⁻¹²⁰⁸ protein, in which we replaced the corresponding Lys/Arg residues to alanine (and in one case to a glutamate) (see Fig. 6a) and analyzed these variants for DNA binding and helicase activity (Fig. 6b-d). The results confirm that the lower WH domain of the R4ZBD is not involved in the helicase mechanism, as DNA binding and helicase activity was indistinguishable from the wild-type protein. Therefore, we concluded that the lower WH domain rather represents a protein-protein interaction module or, alternatively, recognizes more complex DNA metabolism intermediate structures, which are important for RecQ4 *in vivo* activity.

2. It is possible to delete either the entire R4ZBD domain or upper WH domain alone (for example by replacing the domain with a short flexible linker sequence)? If it is

indeed a protein-protein interaction module as suggested it may be dispensable for the helicase activity of the protein.

In line with the above mentioned suggestion for the isolated R4ZBD, we cloned and expressed two deletion variants of the RecQ4⁴²⁷⁻¹²⁰⁸ protein, which affected the entire R4ZBD. Unfortunately, deletion of the complete R4ZBD resulted in only very small amounts of soluble protein and further purification was problematic as all R4ZBD deletion variants aggregated upon concentration. In contrast, deletion of the upper WH domain (Δ 944-1031) was possible. The biochemical analysis of the Δ 944-1031 variant was performed in parallel with the lower-WH domain mutational variants and the results demonstrate that the entire upper WH domain is dispensable for DNA binding and helicase activity (Fig. 6b-d). We only observed a somewhat reduced velocity of helicase activity but the plateau of helicase activity is comparable to the WT protein. Furthermore, we observed a wild-type like DNA binding affinity of this deletion variant. Consequently, we conclude that the upper WH domain is not directly involved in the helicase mechanism.

3. By analogy to RQC domains of other RecQ proteins that remain in the same position regardless of the presence or absence of DNA, the authors claim that the observed position of the R4ZBD in hsRecQ4 is not simply due to lack of DNA substrate (lines 185-188). I don't really think is a strong enough argument - after all, they are structurally different domains that likely have differing functions. Also, large domain motions have been observed in other DNA helicases upon substrate binding. Testing this (perhaps by SAXS or FRET studies) would substantially enhance the study. Alternatively, this claim should be dropped.

We appreciate the comment and acknowledge the need for supporting data for this particular claim. We removed this argument from the manuscript.

4. The authors carry out a helicase activity assay of a C-truncated construct (427-1116) as well as one including the C-terminus (427-1208) and show that longer version as about 5-fold more active than the truncation (figures 5a and 5b). They conclude that the shorter construct can support helicase activity (lines 220-221). It would be interesting if the authors could include in their assay a known helicase-defective point mutant (for example in one of the signature helicase motifs) to allow

better assessment of the absolute activity of the shorter protein.

We included helicase data for two ATPase impaired WalkerA/B mutants (K508A and D605A) for the RecQ4⁴²⁷⁻¹²⁰⁸ protein variant in Fig. 5a and b and show that they are inactive.

Minor Points:

1. In the introduction and abstract it is stated that RecQ4 lacks characteristic RecQ domains. How then is it defined as a member of the RecQ family?

Affiliation to the family of RecQ helicases is based on the homology of the ATPase domain towards the RecQ-family founding member in *Escherichia coli*. We have included a corresponding statement in the introduction of the revised manuscript.

2. Mutations in RecQ4 are linked to several diseases (introduction, lines 58-59). It would be useful if the authors could summarise the phenotypes of these syndromes, so that the reader might better appreciate the physiological effects of RecQ impairment.

We have included a short paragraph about the phenotypes of RecQ4 associated diseases in the introduction of the revised manuscript.

3. It is not really clear what is meant by the use of the word “topologically” (line 138). This is also confusing as the authors later claim that the R4ZBD domain cannot be separated into two units (lines 144-145).

We apologize for the confusing statements. As the overall description of our structure appears in the text prior to the paragraphs where we compare our structure to other RecQs and where we compare the R4ZBD with two distinct types of WH domains, we wanted to point out the structural differences between the R4ZBD and the RQC domain. While the RQC domain consists of two clearly separated autonomous subdomains, the top WH domain in RecQ4 represents an insertion within the lower WH domain and the entire R4ZBD can thus not be described as a domain of two individually folding units. We have changed the corresponding paragraph accordingly

to avoid any confusion.

4. How do the authors know that the R4ZBD is packed “loosely” against HD1 (lines 152-153). Presumably if there are salt bridges and hydrogen bonds between the domains, they are not loosely bound!

A domain interaction analysis performed by the PISA server highlighted several residues, which mediate interactions between the R4ZBD and the HD1, however, the strength of this interaction was rated very low, which led to our description of the interaction as a “loosely” packed arrangement. As we do not mention or discuss the results of this analysis, however, we have rephrased this paragraph in the revised manuscript accordingly.

5. Line 199. none-Zn should be non-Zn I think.

We have changed this accordingly.

6. Labelling locations of the N- and C-termini as well as sequence numbers at the positions of the domain boundaries in figure 1b would aid reader comprehension.

We have updated Fig. 1b and highlighted the location of the N- and C-terminus. However, we would like to refrain from any further labeling of domain boundaries in Fig. 1b as we fear that this would overcomplicate the Figure and might cause confusion. As an alternative, we have included additional domain boundaries in the RecQ4 illustration of Fig. 1a and provide a comprehensive annotation of secondary structure elements in supplementary figure 1.

Reviewer #2 (Remarks to the Author):

The paper “Human RecQ4 structure and function reveals bacterial ancestry; a potential target in cancer therapy” by Kaiser et al presents the crystal structure of human RecQ4 helicase ATPase core domain with a unique C-terminal extension domain as well as biochemical characterization of several mutants. Experimental data are solid and the structure of unique R4ZBD domain is very interesting. Unfortunately, the properties and function of novel domain were not addressed

experimentally, main conclusions are very speculative and, as a result, the paper present only incremental advance in the field. Therefore, it is not of interest for a wide research community and has to be published in more specialized journal.

Specific concerns:

1) The authors did not study functional property of novel structural domain, which significantly deviates from RQC domain mentioned by the authors as a part of functional core in all other RecQ helicases. Instead they targeted 1) the C-terminal extension with unknown structure, and 2) ARL motif already studied in other helicase. The former resulted in additional role of C-terminal domain for DNA binding, although whether it is due to additional DNA-binding region or structural role for core stability is unknown. The later just confirmed identical function of ARL motif from other helicase. Thus, although experiments were properly conducted, results are not significant. It is unclear why the authors dedicated majority of the paper to structural comparison of the novel R4ZBD domain with known structures, but did not attempt to test its functionality biochemically.

Reviewer #1 expressed a similar concern regarding the functional characterization of the R4ZBD and we agree that a functional analysis would support some of the claims of our manuscript. Accordingly, we pursued a mutational and functional analysis of the DNA binding properties and helicase activity of the R4ZBD and included the results in our revised manuscript (Fig. 6). Generally, the data support our suggestion that the R4ZBD is not involved in DNA binding or dsDNA strand separation of a 3'OH DNA substrate and might thus rather represent a protein-protein interaction module. This emphasizes the importance of the CTD as it represents the major determinant, which facilitates dsDNA strand separation. The analysis of the ARL in RecQ4 is required to confirm a conserved ssDNA binding- and translocation-mechanism between RecQ4 and *EcRecQ*, which allows us to draw conclusions about the trajectory of the DNA substrate and provides, for the first time, functional insights into the structure-function relationship of individual ARL residues.

2) The choice of the title is odd and some conclusions are highly speculative and not supported by experimental data.

First, bacterial heritage does not warrant drug design advantage to target specific

protein. Any structural differences can be used for inhibition specificity. Moreover, the paper only briefly refers to rational of RecQ4 targeting for drug design without addressing it in more details. This is not a topic of current research and should not be in a title of the paper.

We appreciate the comment and agree that the title we have chosen might misguide the expectations of what would be presented and discussed in the paper. We chose a more appropriate title for the revised manuscript.

Second, the conclusion about protein similarities are based on the absence of the b-hairpin in RecQ4 combined with similarities in ARL motif leading to proposal that RecQ4 should bent DNA during unwinding to a larger degree than other human RecQs and similarly to that of the bacterial RecQ (what is the difference in bending degree is not discussed). This logic is odd at multiple levels. Only small portion of novel domain overlaps spatially with RQC and large structural differences point to differences in DNA binding between RecQ4 and remaining RecQs with the conserved RQC domain, rather than similarities. Note, that bacterial homolog does contain b-hairpin, even though its mutation does not affect unwinding to the same degree as in other helicases.

It is also unclear why similar function of RecQ4 ARL motif with the one in bacterial RecQ distinguish it from other human RecQs? Are there known differences in properties of ARL motifs?

We agree with the concerns, expressed by Reviewer #2 that our discussion of the conclusions regarding the possible helicase mechanism has favored a bend-angle driven mechanism over a physical wedge element. We have rewritten a large proportion of the discussion regarding this issue in order to clarify that neither of the two different helicase mechanisms can be fully excluded at this point. However, the absence of the CTD in RecQ4⁴²⁷⁻¹¹¹⁶ does permit approximately 20% of wild-type helicase activity. In our view, this result is more compatible with a bend-angle driven mechanism, as “critical” ssDNA/dsDNA conformations are still possible via random thermal motions of the DNA substrate while the removal of a physical wedge element would probably abolish helicase activity completely. However, we agree that there are additional experiments required to fully confirm one particular helicase mechanism and we have thus adapted our discussion accordingly.

Regarding our analysis concerning the ARL, we apologize if we have not made our rationale for this analysis clear enough. The *E. coli* RecQ helicase is the only RecQ helicase where the ARL was analyzed comprehensively and to a full extent (Manthei *et al.*, PNAS, 2015). As this analysis permits conclusions about the trajectory of the DNA substrate, we chose to analyze the ARL in RecQ4 in order to demonstrate that the ssDNA binding- and translocation-mechanism is conserved between human RecQ4 and *E. coli* RecQ, and thus by further extend, also with the other human RecQ helicases. Based on these data, we concluded that the dsDNA entity of the DNA substrate should be located in close proximity to the RecQ4 CTD, which is supported by our functional analysis of the R4ZBD and the fact that the CTD is important for RecQ4 helicase activity. We have adapted the discussion of our revised manuscript regarding the points mentioned above in order to explain the rationale of the ARL analysis and its results. Finally, we want to point out that our results of the RecQ4 ARL allowed us to correlate reduced ATPase activity with the grade of helicase activity reduction. Thus, our ARL analysis permits for the first time a detailed structure-function correlation of individual ARL residues and might serve as a standard example for a comprehensive ARL analysis in other RecQ helicases.

Minor points:

1) It will be helpful to have one structural picture with numbers of important residues in main text or at least the sequence with secondary structure elements and numbering. It is difficult for the reader to follow description and discussion otherwise.

Reviewer #1 expressed a similar concern. In order to help the reader to follow the description in the results and discussion, we have updated Fig. 1b and included the location of the N- and C-terminus. Additionally, we have added all informative domain boundaries in the RecQ4-illustration of Fig. 1a and provide a comprehensive annotation of secondary structure elements, maintaining the domain color code as shown in Fig. 1a and 1b, in Supplementary Fig. 1. Both, Fig. 1b and Supplementary Fig. 1 highlight the location of the ARL and Fig. 6a summarizes the location of the R4ZBD mutations. However, we would like to refrain from any further labeling of domain boundaries in Fig. 1b as we fear that this would overcomplicate the Figure and might cause confusion.

2) Spell HD1,2 abbreviation at the beginning.

We have changed this accordingly.

3) Lane 113: "rore"

We have changed this accordingly.

4) The significance of similar location of helices with zinc finger of RQC is unclear. This region does not form contact dsDNA and is a linker between HD2 and RQC, location of which is very different from that of R4ZBD. If R4ZBD binds DNA, then it should either swivel around α -helices or support very different conformation of DNA.

We apologize for the confusion, which we might have caused by our structural analysis. Reviewer #2 correctly states that the R4ZBD Zn^{2+} -binding site and the RQC-mediated Zn^{2+} -binding domain occupy very different locations. However, our structural analysis focuses on the fact that the location of the conserved Zn^{2+} -binding domain in *EcRecQ* and *HsRecQ1*, is exactly occupied by RecQ4s bridging helix α 19, bridging helix α 11 and helix α 20 (see Fig. 4). This arrangement in RecQ4 does of course not coordinate a Zn^{2+} ion, as the zinc coordinating residues of RecQ4 are located in the R4ZBD. We do not correlate or indicate any DNA interactions based on this structural similarity. However, we hypothesize that the RQC- Zn^{2+} -binding domains and the similar arrangement within RecQ4 (mediated by the RecQ4 bridging helices and helix α 20) could be important for structural stability of the protein. This hypothesis is supported by the analysis of two RecQ4 disease mutations, which prematurely terminate the protein within and after helix α 19 and are associated with RAPADILINO syndrome. Furthermore, we could not achieve soluble expression of a RecQ4⁴²⁷⁻¹⁰⁹⁰ variant, which represents one of the disease mutations, although the only difference to the soluble RecQ4⁴²⁷⁻¹¹¹⁶ variant is the absence of helix α 20, which further supports our hypothesis.

We included a statement in our current manuscript that RecQ4 does not coordinate a zinc ion at the position described above in order to make this point more clear.

5) Discussion is extremely lengthy, repetitive and excessively speculative, even for structural paper.

During the revision of our manuscript, we have paid special attention on shortening our explanations and lengthy conclusions and deleting any repetitive statements within the discussion section. Although we had to incorporate and discuss the new results, which were requested by the Reviewers, we were able to shorten the discussion from approximately 2100 words to a final word count of approximately 1750.

REVIEWERS' COMMENTS:

Reviewer #1 (Remarks to the Author):

I'm satisfied that the authors have addressed all the issues raised in my original review, and would support publication of the revised manuscript.

Reviewer #2 (Remarks to the Author):

In the revised manuscript, the authors added mutagenesis studies of novel structural domain, which ruled out its involvement in DNA binding. Overall, the manuscript 1) presents the structure of the core helicase domain of RecQ4, 2) provides description of solution properties based on traditional biochemical DNA binding, ATPase and unwinding assays, 3) describes the structure and mutagenesis of the novel R4ZBD domain uniquely present in RecQ4 instead of the conserved RQC domain, 4) rules out DNA-binding properties of R4ZBD domain, and 5) reveals the importance of the C-terminal residues (unresolved in structure) for helicase activity.

Studies are well performed (even though several conclusions are rather speculative or oversimplified) and the novel structure will definitely play an important role in unmasking mystery of functional specificity and mechanism of RecQ helicases in future. However, the results as presented do not provide significantly new mechanistic insight nor lead to a novel hypothesis about function and mechanism of RecQ4. Therefore, this reviewer will not recommend the manuscript for publication in Nature Communication.